# Potential Therapeutic Agents against Paclitaxel—And Sorafenib-Resistant Papillary Thyroid Carcinoma

**DOI:** 10.3390/ijms231810378

**Published:** 2022-09-08

**Authors:** Seok-Mo Kim, Keunwan Park, Jin Hong Lim, Hyeok Jun Yun, Sang Yong Kim, Kyung Hwa Choi, Chan Wung Kim, Jae Ha Lee, Raymond Weicker, Cheol-Ho Pan, Ki Cheong Park

**Affiliations:** 1Department of Surgery, Thyroid Cancer Center, Gangnam Severance Hospital, Institute of Refractory Thyroid Cancer, Yonsei University College of Medicine, Seoul 06273, Korea; 2Natural Product Informatics Research Center, KIST Gangneung Institute of Natural Products, Gangneung 25451, Korea; 3Department of Surgery, Gangnam Severance Hospital, Yonsei University College of Medicine, 211 Eonjuro, Gangnam-gu, Seoul 06273, Korea; 4Department of Surgery, Yonsei University College of Medicine, 50-1, Yonsei-ro, Seodaemun-gu, Seoul 03722, Korea; 5Department of Urology, CHA Bundang Medical Center, CHA University, Seongnam 13496, Korea; 6CKP Therapeutics, Inc., 110 Canal Street, 4th Floor, Lowell, MA 01852, USA

**Keywords:** patient-derived xenograft tumor model, paclitaxel, sorafenib, drug-resistant papillary thyroid carcinoma

## Abstract

Thyroid carcinoma, a disease in which malignant cells form in the thyroid tissue, is the most common endocrine carcinoma, with papillary thyroid carcinoma (PTC) accounting for nearly 80% of total thyroid carcinoma cases. However, the management of metastatic or recurrent therapy-refractory PTC is challenging and requires complex carcinoma therapy. In this study, we proposed a new clinical approach for the treatment of therapy-refractory PTC. We identified sarco/endoplasmic reticulum calcium ATPase (SERCA) as an essential factor for the survival of PTC cells refractory to the treatment with paclitaxel or sorafenib. We validated its use as a potential target for developing drugs against resistant PTC, by using patient-derived paclitaxel- or sorafenib-resistant PTC cells. We further discovered novel SERCA inhibitors, candidates 7 and 13, using the evolutionary chemical binding similarity method. These novel SERCA inhibitors determined a substantial reduction of tumors in a patient-derived xenograft tumor model developed using paclitaxel- or sorafenib-resistant PTC cells. These results could provide a basis for clinically meaningful progress in the treatment of refractory PTC by identifying a novel therapeutic strategy: using a combination therapy between sorafenib or paclitaxel and specific SERCA inhibitors for effectively and selectively targeting extremely malignant cells such as antineoplastic-resistant and carcinoma stem-like cells.

## 1. Introduction

Thyroid carcinoma represents more than 90% of all endocrine cancers, and its incidence has raised in the past four decades [1]. It is categorized into four major types: papillary (PTC), follicular (FTC), medullary (MTC), and anaplastic thyroid carcinoma (ATC) [2,3]. Additionally, thyroid carcinoma is frequently classified in accordance with the immunohistochemical and molecular characteristics of tumor cells in differentiated or undifferentiated [4,5]. Well-differentiated thyroid carcinoma (WDTC) generally has a good prognosis and higher treatment response rates than poorly differentiated (PDTC) or undifferentiated thyroid carcinoma (UTC), which is rarer, aggressive, associated with early metastasis, and has a very poor prognosis [6,7,8]. Although PTC is a type of WDTC, it often develops antineoplastic resistance, which is a critical issue as it promotes death by metastasis or cancer recurrence [9,10]. Antineoplastic-sensitive and -resistant cancers have different biological and molecular characteristics and distinct clinical behaviors [11,12]. Various studies have indicated that these two types of cancer are also characterized by different mutations [13,14,15]. Several researches have revealed that the FGFR (fibroblast growth factor receptor) signaling pathway plays a conclusive part in EMT (epithelial–mesenchymal transition)-associated drug resistance and poor prognosis of refractory cancer by its change of cancer stemness or aggressiveness of refractory cancer cells [16,17]. Late study trends have shown that an advance of EMT in refractory cancer cells not only affects outcomes in drug resistance, but is also a crucial leading factor in metastasis via the FGFR signaling pathway [18,19]. Although extensive efforts have been undertaken to identify the molecular mechanisms underlying the poor prognosis of patients with antineoplastic-resistant PTC, no research has yet provided a definite explanation. Furthermore, refractory PTC is characterized by continually gaining drug resistance. For this reason, effective clinical approaches for managing this disease are very much needed [10,20,21]. The present study identified the sarco/endoplasmic reticulum calcium ATPase (SERCA) as being dominantly expressed in a patient-derived xenograft tumor model developed with antineoplastic-resistant PTC cells, and not in one developed using antineoplastic-sensitive PTC cells. SERCA is well-known as an essential regulator of cytosolic free calcium [22,23,24] and, consequently, of many cellular processes, including cellular survival or death by apoptosis, and autophagy in case of severe endoplasmic reticulum (ER) stress [24,25,26]. Furthermore, we validated SERCA as a target of potential therapeutic agents for the treatment of antineoplastic-resistant PTC and identified compounds 7 and 13 as novel SERCA inhibitors.

Our discoveries might have a substantial clinical impact: by using innovative target-focused combinatorial strategies, we were able to selectively and efficaciously target highly malignant cells, such as drug-resistant cancer cells.

## 2. Results

### 2.1. Patient Disease Characteristics

In total, 21 medical records of patients with advanced or metastatic PTC consecutively treated with paclitaxel (*n* = 5) or sorafenib (*n* = 16) at our center were analyzed. Patient disease characteristics and demographics by treatment agent are presented in Figure 1A. The mean age of the patients was 58.0 ± 11.3 years, and 71.4% of patients were females. Distant metastases were observed in 76.2% of patients, and locally advanced PTC was identified in 23.8% of patients. The cases of distant metastatic lesions in the lungs were 40.0% and 87.5% in the paclitaxel and sorafenib groups, respectively, and 20.0% and 25.0% in the bone, respectively. All patients had received previous RAI, and the mean cumulative radioactive iodine (RAI) was 442.0 ± 241.3 mCi among those treated with paclitaxel, and 657.5 ± 409.2 mCi among those treated with sorafenib. External beam radiation therapy was performed in 40.0% and 62.5% of the paclitaxel and sorafenib groups. The overall survival of 21 patients was 133.9 ± 62.5 months, but there was no significant difference between the paclitaxel and sorafenib groups (Figure 1B). Overall survival for 21 patients was 133.9 ± 62.5 months. The patient survival rate was 90.2% at 5 years and 84.5% at 10 years after detection of thyroid carcinoma (Figure 1C). 

### 2.2. Characteristics of Patient-Derived Drug-Resistant PTC Cell Lines

In this study, various PTC cell lines were generated from tumor specimens collected from patients (Figure 2A); YUMC-S-P1 (the first cell line developed from patient-derived antineoplastic-sensitive PTC cells), YUMC-R-P1, -P2, and -P3 (first, second, and third cell lines developed from patient-derived antineoplastic-resistant PTC cells) cell lines were developed from tumors of patients who received PTC treatment at Severance Hospital, Yonsei University College of Medicine, Seoul, Republic of Korea. YUMC-R-P1 cells resistant to paclitaxel, and -P2 and -P3 cells resistant to sorafenib were more aggressive (metastasis or recurrent) than antineoplastic-sensitive PTC cells, YUMC-S-P1, and patients from whom the former cell lines were developed, presented recurrence or metastasis (Figure 2A). To evaluate and compare the genetic differences and signaling pathways activated in both forms of PTC by using YUMC-S-P1 and YUMC-R-P1, -P2, and -P3 cells, we carried out an RNA-sequencing (RNA-Seq)-mediated transcriptome analysis (Figure 2B–F). 

RNA-Seq analysis indicated that YUMC-R-P1, -P2, and -P3 cells were associated with critically higher levels of the expression of epithelial-mesenchymal transition (EMT) markers *SNAIL* (Zinc finger protein SNAI1), *ZEB* (Zinc finger E-box-binding homeobox), and *TWIST* (twist family bHLH transcription factor) when compared with YUMC-S-P1, as illustrated in Figure 2B. In paclitaxel- or sorafenib-resistant PTC, besides *TWIST*, *ZEB*, and *SNAIL*, *FGF* (Fibroblast growth factor) and *FGFR* are also highly expressed (Figure 2B). Cancer stem cell- (CSC, top), FGF- (middle) and EMT (bottom)-associated genes exhibited higher expression levels in paclitaxel- or sorafenib-resistant than in antineoplastic-sensitive PTC (Figure 2C). Kyoto Encyclopedia of Genes and Genomes (KEGG) pathway analysis revealed enrichment in cancer stemness-related signaling pathways (Hedgehog, calcium, Wnt, PPAR, TGF/SMAD, and PI3K/Akt signaling pathway) [27] in resistant PTC when compared to antineoplastic-sensitive PTC (Figure 2D–F). Furthermore, the calcium signaling pathways were also highly enhanced in paclitaxel- or sorafenib-resistant PTC cells (Figure 2D–F). Furthermore, basal levels of the protein expression of SERCA, which is a critical player in apoptosis and calcium homeostasis [23,28,29,30], were higher in YUMC-R-P1, -P2, and -P3 than in YUMC-S-P1 cells (Figure 2G). Moreover, after exposure to paclitaxel or sorafenib, the levels of expression of SERCA significantly increased in YUMC-R-P1, -P2, and -P3 compared to YUMC-S-P1 cells (Figure 2G).

Our results indicate gene pathways related to cancer stemness and SERCA expressions are essential for the survival of resistant PTC cells to antineoplastic therapy. Therefore, they can be important targets for developing new anticancer drugs against recurrent and metastatic forms of drug-resistant PTC. However, their efficacy must be assessed in clinical trials.

### 2.3. Discovery of Candidates 7 and 13 as Specific Inhibitors of SERCA Using In Silico Screening

We hypothesized that the functional inhibition of SERCA might be an effective clinical approach for treating paclitaxel- and sorafenib-resistant PTC. We searched for SERCA-binding compounds and used in silico screening and pharmacophore modelling to analyze their binding modes to the target. For screening, we used a novel evolutionary chemical binding similarity (ECBS) method based on a classification similarity-learning framework defined with paired chemical data and the target’s evolutionary relationship. Consequently, 1423 (assessed by using the docking score), 67 (selected manually), and eventually 27 candidate compounds (that included natural compounds) were identified. Of these, 27 candidate compounds had high binding affinity for SERCA (Figure 3A, top). Natural origin compounds 7 and 13 were identified as highly effective inhibitors of SERCA (Figure 3A, bottom), and were selected for further studies (Figure 3B). Candidates 7 and 13 could provide original therapeutic strategies against antineoplastic-resistant PTC. 

### 2.4. Inhibitory Effect of Novel Candidates 7 and 13 on SERCA and Their Effect on The Survival of Paclitaxel- and Sorafenib-Resistant PTC Cells

To evaluate the anticancer efficacy of candidates 7 and 13, we carried out cell viability assays for YUMC-S-P1 and YUMC-R-P1, -P2, and -P3 cells treated with either paclitaxel or sorafenib, or with a combination between paclitaxel or sorafenib and candidates 7 or 13. The viability of YUMC-S-P1 significantly decreased in a dose-dependent manner, following exposure to paclitaxel and sorafenib alone or in combination with candidates 7 or 13 (Figure 4A, top). The viabilities of YUMC-R-P1, -P2, and -P3 cells were not influenced by paclitaxel or sorafenib treatment, respectively. In contrast, combinations of paclitaxel or sorafenib with thapsigargin, a known SERCA inhibitor (positive control) and candidates 7 or 13, the newly identified SERCA inhibitors, reduced the viability of paclitaxel- or sorafenib-resistant PTC cells substantially and in a dose-dependent manner (Figure 4A, bottom). Furthermore, exposure to a single SERCA inhibitor, thapsigargin, and candidates 7 or 13 did not substantially influence the viability of antineoplastic-sensitive and -resistant PTC cells. In antineoplastic-sensitive PTC cells, the half-maximal inhibitory concentration (IC_50_) was 21.3 nM for paclitaxel and 25.5 uM for sorafenib, respectively. Furthermore, IC_50_ values of paclitaxel and sorafenib did not change significantly when they were associated with thapsigargin or the novel inhibitors. Conversely, in paclitaxel or sorafenib-resistant PTC cells, paclitaxel or sorafenib treatment alone had minimal anticancer effect. However, their anticancer effect was substantially enhanced by the combination with SERCA inhibitors. When combined with these SERCA inhibitors, paclitaxel had an IC_50_ of 14 nM in YUMC-R-P1 cells, while sorafenib had an IC_50_ of 20–26 uM in YUMC-R-P2 and -P3 cells (Figure 4B). 

These results indicate that SERCA is an essential factor for the prolonged survival of drug-resistant cancer cells by restoring the levels of cytosolic free calcium and preventing its overload.

### 2.5. The Effect of the Co-Treatment of Paclitaxel or Sorafenib and SERCA Inhibitors on ER Stress-Mediated Apoptosis in YUMC-R-P1, -P2, and -P3 Cells

We carried out flow cytometry assay and immunoblot analysis to assess the mechanism underlying the effects of paclitaxel or sorafenib alone, or those of the co-treatment with paclitaxel or sorafenib and candidates 7 and 13 on YUMC-R-P1, -P2, and -P3 cells. Thapsigargin was used as positive control. All treatments determined a substantial increase in the level of expression of sub-G_0_/G_1_ in YUMC-S-P1 cells. This induced tumor cell apoptosis in paclitaxel- or sorafenib-treated groups with or without SERCA inhibitors (Figure 5A, left). Moreover, YUMC-S-P1 cells also showed higher levels of expression of CHOP (marker of ER stress), cytosolic cytochrome c, and cleaved caspase-3, which also promote apoptosis (Figure 5A, right). This may explain the anticancer efficacy of paclitaxel or sorafenib as singular therapeutic agents in antineoplastic-sensitive PTC. In contrast, in paclitaxel- or sorafenib-resistant PTC cells, monotherapy with paclitaxel or sorafenib did not significantly influence the level of sub-G_0_/G_1_ (Figure 5B–D, left). However, when combined with SERCA inhibitors, they substantially increased the level of sub-G_0_/G_1_, thus resulting in induction of apoptosis (Figure 5B–D, left). Immunoblot assay results indicated that monotherapy with paclitaxel or sorafenib did not influence ER stress-mediated apoptosis in paclitaxel- or sorafenib-resistant PTC cells (Figure 5B–D, right). However, combined treatment with SERCA inhibitors highly elevated the ER stress-mediated apoptosis, as highlighted by the increase of the levels of CHOP (marker of ER stress), cytosolic cytochrome c, and cleaved caspase-3 (Figure 5B–D, right). 

These results demonstrate that in paclitaxel- or sorafenib-resistant PTC cells, combined treatment with paclitaxel or sorafenib and SERCA inhibitors promotes apoptosis by inducing ER stress and activating cytochrome *c*-dependent pathways.

### 2.6. In Vivo Assessment of the Anticancer Efficacy of Candidates 7 and 13 in a Patient-Derived Drug-Resistant PTC Cell Mouse Xenograft Model

We developed mouse xenograft models using antineoplastic-sensitive (YUMC-S-P1) and -resistant PTC (YUMC-R-P1, -P2, and -P3) cells and evaluated the anticancer effect of monotherapy with paclitaxel, sorafenib, or with SERCA inhibitors (thapsigargin, candidates 7 and 13) as well as that of the combination therapy between paclitaxel or sorafenib and a SERCA inhibitor. In the xenograft model developed using antineoplastic sensitive PTC cells, paclitaxel or sorafenib administration with or without SERCA inhibitors significantly decreased tumor size (Figure 6A, left, top and bottom). No significant changes in resected tumor weight were found between any of the treatment groups (Figure 6A, right, top and bottom). None of the therapy regimens did significantly influence mouse body weight (Figure 6A, middle, top and bottom). Nonetheless, in the xenograft model developed using paclitaxel- or sorafenib-resistant PTC cells, monotherapy with either paclitaxel or sorafenib did not induce significant tumor shrinkage (Figure 6B–D, left, top: YUMC-RP1; middle: -P2; bottom: -P3). However, combination with SERCA inhibitors resulted in substantial tumor shrinkage. Similar results were observed for the weight of the resected tumor (Figure 6B–D, right, top: YUMC-RP1; middle: -P2; bottom: -P3). None of the treatment regimens did significantly influence mouse body weight (Figure 6B-D, middle, top: YUMC-RP1; middle: -P2; bottom: -P3). 

Taken together, these results showed that novel SERCA inhibitors, candidates 7 and 13, could provide a novel clinical approach for treating patients with refractory PTC, as they induce significant tumor shrinkage in a xenograft tumor model developed using patient-derived antineoplastic-resistant cells.

## 3. Discussion

PTC is a frequent endocrine carcinoma that often has a good prognosis [31,32] and a good response rate to therapy. In contrast, antineoplastic-resistant PTC shows a poor prognosis as it may result in metastases or recurrence, outcomes frequently associated with death. Recurrent or metastatic PTC is refractory to most medical therapy [33,34]. Numerous genes were proposed between anticancer drug-sensitive and -resistant PTC, exposing that EMT was the acute factor in cancer aggressiveness and stemness [35]. In this study, EMT-related genes were more extremely increased in paclitaxel or sorafenib-resistant PTC when compared with that in paclitaxel or sorafenib-sensitive PTC. Previous well-known researches showed that EMT of cancer is identified as malignant progression such as invasion, metastasis, and anticancer drug resistance [36,37]. In particular, CSCs (cancer stem cells) that showed EMT were considered to be decisive for metastasis, recurrent, and drug resistant, as has been revealed in numerous human anticancer drug-refractory cancers [38,39]. Some researches have also proved the connection between drug resistance and EMT in CSCs [39,40]. A notable point in this study was that paclitaxel- or sorafenib-resistant PTC had properties similar to CSCs in terms of EMT-involved stemness. Various cytogenetic events and oncogenic mechanisms are involved in the oncogenesis of refractory thyroid carcinoma [41,42]. Rearrangement of the RET proto-oncogene is a specific genetic change monitored in PTC, but not in UTC [41]. To date, the aggressiveness and cancer stemness of refractory PTC has not been fully explained.

Anticancer drug discovery has evolved and new therapeutic strategies against cancer have emerged. Thus, preoperative chemotherapy expands survival rates after surgery, and adequate chemotherapy and surgery are effective even in unfavorable clinical situations [43,44,45,46]. However, there are no effective therapeutic options for the management of antineoplastic-resistant cancer [47,48]. Consequently, it has high mortality. Recurrent and metastatic antineoplastic-refractory cancer is a critical unmet medical need [49,50]. In refractory cancer, destroying drug-resistant tumor cells is a primary objective of the therapy [51,52]. Thus, management of anti-neoplastic refractory cancer represents the major challenge in cancer treatment [52,53]. Depending on the type of cancer and its specific characteristics, resistance to antineoplastics may be acquired by different mechanisms and leads to the death of the patients [54]. Consequently, a therapeutic solution for antineoplastic-resistant cancer might seem an unreachable goal.

In this research, we used mRNA-Seq analysis of patient-derived antineoplastic-resistant PTC to discover targets for developing specific anticancer inhibitors. Using ECBS, in silico screening, and patient-derived paclitaxel- or sorafenib-resistant PTC cells, we identified selective inhibitors of SERCA which can be used to build a therapeutical scheme for antineoplastic-resistant cancer. We carried out KEGG pathway analysis, and demonstrated that calcium and hedgehog signaling pathways are among the top 15 highly enhanced signaling pathways in paclitaxel- or sorafenib-resistant, but not in anti-neoplastic-sensitive, PTC cells. Furthermore, mRNA-Seq analysis disclosed that the level of expression of SERCA is high in paclitaxel- or sorafenib-resistant PTC cells. Previous research has demonstrated the correlation between Notch and calcium signaling pathways [55,56]. Furthermore, SERCA inhibition suppresses Notch1 signaling [57,58]. Therefore, we focused primarily on the role which critical signaling pathways and genes related to calcium homeostasis and cell survival play in paclitaxel- or sorafenib-resistant PTC cells. The primary objective of developing anticancer drug is to identify therapeutical agents which kill cancer cells. Despite the well-known anticancer efficacy of paclitaxel and sorafenib in various types of cancer, tumors may acquire resistance to these agents, which thus become ineffective. In our previous article, we revealed that under metabolic stress conditions, metabolic stress-resistant cancer cells escape apoptosis induced by cytosolic free calcium overload by SERCA induction [59]. Our current research demonstrates that in paclitaxel- or sorafenib-resistant PTC cells, suppression of SERCA activity using novel specific inhibitors, such as candidates 7 or 13, leads to apoptosis by inducing severe ER stress. This effect is independent of the extent of the increase in the levels of expression of SERCA.

The work presented in this article proved that in patient-derived paclitaxel- or sorafenib-resistant PTC cells, SERCA is a critical regulator of the apoptosis induced by the overload of cytosolic free calcium under anticancer drug treatment. Based on these findings, SERCA might be a novel therapeutic target present in drug-resistant PTC cells, and its activation might be one of the most efficacious mechanisms underlying the prolonged survival of drug-resistant PTC cells under antineoplastic treatment. Furthermore, in this study, two novel SERCA inhibitors, candidates 7 and 13, destroyed antineoplastic-resistant PTC cells exposed to severe ER stress, such as that induced by antineoplastic treatment.

Our findings might provide a basis for planning future logical and effective therapeutic approaches for patients with refractory PTC. For this purpose, we propose a therapeutic approach based on the genetic differences between antineoplastic-resistant and sensitive PTC.

Further research must be undertaken to ensure the results of this study may be translated to clinical settings.

## 4. Materials and Methods

### 4.1. Study Design and Ethical Considerations

This study was a retrospective, single-center analysis of patients diagnosed with PTC (between January 2003 and December 2019), as detailed in our previous study. All procedures involving patients were performed in accordance with the institutional ethical standards, all applicable local/national regulations, and guidelines of the 1964 Helsinki Declaration and its later amendments. In accordance with the Bioethics and Safety Act of Korea, formal written consent was not required for this type of retrospective, observational analysis. The study protocol was approved by the Institutional Review Board (IRB) of Severance Hospital, Yonsei University College of Medicine (IRB protocol: 3-2019-0281). Cell samples were obtained from patients at the Severance Hospital, Yonsei University College of Medicine, Seoul, Korea.

### 4.2. Patients

#### 4.2.1. Patient 1

YUMC-S-P1 was 31-year-old woman with papillary thyroid carcinoma. This patient had bilateral thyroid tumors with extrathyroidal extension. This woman underwent bilateral total thyroidectomy and bilateral modified radical neck dissection with central compartment neck dissection. Surgical findings showed that the tumor invaded the recurrent laryngeal nerve and was removed by careful shaving. After surgery, she was given high-dose radioiodine ablation therapy 3 times. Currently, radiologic examination and thyroid hormone tests are being followed without recurrence. The pathology report indicates the presence of papillary thyroid carcinoma with both lateral metastatic lymph nodes.

#### 4.2.2. Patient 2

YUMC-R-P1 was 52-year-old woman with papillary thyroid carcinoma. This patient had multiple tumors and extensive extrathyroidal extension. This woman underwent bilateral total thyroidectomy with central compartment neck dissection. One year after surgery, metastasis to the mediastinum and right lateral cervical lymph nodes was confirmed, and she underwent mediastinal dissection through partial sternotomy and right modified radical neck dissection. The specimens for culture were obtained after the last operation. This patient underwent paclitaxel, after which the disease progression was confirmed in the anticancer drug response evaluation. Cancer recurrence and metastasis were caused after paclitaxel prescribed.

#### 4.2.3. Patient 3

YUMC-R-P2 was a 57-year-old man with papillary thyroid carcinoma. After a bilateral total thyroidectomy with central compartment neck resection, this man experienced left radical nephrectomy and right lung wedge resection for kidney and lung metastasis. Afterward, he underwent a right modified radical neck dissection, left lateral selective lymph node dissection. In addition, two additional regional lymph node dissections (left level III) were performed, the pathology state revealed the existence of metastatic poorly differentiated thyroid carcinoma. The specimens for culture were gained after the last surgery. After regional lymph node dissection (left level III), This patient underwent sorafenib, after which the disease progression was confirmed in the anticancer drug response evaluation. Cancer recurrence and metastasis were caused after sorafenib prescribed.

#### 4.2.4. Patient 4

YUMC-R-P3 was a 34-year-old woman with papillary thyroid carcinoma. She underwent bilateral total thyroidectomy with central compartment neck dissection and right modified radical neck dissection. One year after surgery, metastasis to the mediastinum was confirmed, and she underwent mediastinal dissection through transcervical approach. Then, metastasis to the upper mediastinum was confirmed, and mediastinal dissection was additionally performed. The specimens for culture were obtained after the second surgery. After surgery, the pathology report indicates the presence of metastatic papillary thyroid carcinoma. This patient underwent sorafenib, after which the disease progression was confirmed in the anticancer drug response evaluation. Cancer recurrence and metastasis were caused after sorafenib prescribed.

### 4.3. Patient Tissue Specimens

Fresh tumors were collected from patients with histologically and biochemical proven PTC who were treated at the Severance Hospital, Yonsei University College of Medicine, Seoul, Korea. Fresh tumors were collected throughout surgical excision of PTC metastatic and primary sites.

### 4.4. Tumor Cell Isolation and Primary Culture

After resection, tumors were kept in phosphate-buffered saline (PBS) with antifungal and antibiotics, and moved to the laboratory. Normal tissue and fat were removed, and the tissues were rinsed with 1× Hank’s Balanced Salt Solution. Tumors were minced in a tube with dissociation medium containing DMEM/F12 with 20% fetal bovine serum supplemented with 1 mg/mL collagenase type IV (Sigma, St. Louis, MO, USA; C5138). Minced and suspended tumor cells were filtered through sterile nylon cell strainers with 70-micron pores (BD Falcon, Franklin Lakes, NJ, USA), rinsed with 50 mL of 1× Hank’s Balanced Salt Solution, and centrifuged at 220 g for 5 min. Cells were resuspended in RPMI-1640 (Hyclone, South Logan, UT, USA) medium with 10% fetal bovine serum (Hyclone) and 2% penicillin/streptomycin solution (Gibco, Grand Island, NY, USA). Cell viability was determined using the trypan blue dye exclusion method.

### 4.5. mRNA-Seq Data

We pre–processed the raw reads from the sequencer to withdraw low quality and adapter sequence before analysis, and aligned the processed reads to the Homo sapiens (GRCh37) with HISAT v2.1.0 [60]. HISAT utilizes two types of indexes for alignment: a global, whole-genome index, and tens of thousands of small local indexes. Both are constructed using the same Burrows–Wheeler transform (BWT) or graph FM index (GFM) as Bowtie2. Due to the use of these efficient data structures and algorithms, HISAT generates spliced alignments several times faster than Bowtie and the widely used BWA. The reference genome sequence of Homo sapiens (GRCh37) and annotation data were downloaded from the National Center for Biotechnology Information (NCBI). Then, transcript assembly of known transcripts was processed using StringTie v2.1.3b [61]. Based on these results, expression abundances of transcripts and genes were calculated as read count or fragments per kilobase of exon per million fragments mapped (FPKM) value per sample. The expression profiles were used for additional analyses, such as of differentially expressed genes (DEGs). In groups with different conditions, differentially expressed genes or transcripts were filtered through statistical hypothesis testing.

### 4.6. Statistical Aanalysis of Gene Expression Level

The relative abundances of genes were measured in read count using StringTie. We performed statistical analyses to find differentially expressed genes using the estimates of abundances for each gene in the samples. Genes with one more than zero read count values in the samples were excluded. To facilitate log2 transformation, 1 was added to each read count value of filtered genes. Filtered data were log2-transformed and subjected to trimmed mean of M-values (TMM) normalization. The statistical significance of the differential expression data was determined using exactTest, edgeR, and fold change, in which the null hypothesis was that no difference exists among groups. False discovery rate (FDR) was controlled by adjusting the p-value using the Benjamini–Hochberg algorithm. For DEG sets, hierarchical clustering analysis was performed using complete linkage and Euclidean distance as a measure of similarity. Gene-enrichment and functional annotation analysis and pathway analysis for a significant gene list were performed based on Gene Ontology and KEGG pathway analyses.

### 4.7. Hierarchical Clustering

Hierarchical clustering analysis was also performed using complete linkage and Euclidean distance as a measure of similarity to display the expression patterns of differentially expressed transcripts, which are satisfied with |fold change| ≥ 2 and independent *t*-test raw *p* < 0.05. All data analysis and visualization of differentially expressed genes was conducted using R 3.5.1 (www.r-project.org, accessed on 15 October 2021).

### 4.8. Cell Culture

The patient-derived PTC cell lines YUMC-S-P1, YUMC-R-P1, -P2, and -P3 were obtained by tumor cell isolation from the patients and cultured in RPMI-1640 medium with 15–20% fetal bovine serum (FBS; authenticated by short tandem repeat profiling/karyotyping/isoenzyme analysis).

### 4.9. Cell Viability Assay

Cell viability was measured using the MTT (3-(4,5-Dimethylthiazol-2-yl)-2,5-Diphenyltetrazolium Bromide) assay; cells were seeded in 96-well plates at 7 × 10^3^ cells per well and incubated overnight to achieve 80% confluency. The detailed protocol can be found in our previous article [30]. Data were expressed as a percentage of the signal observed in vehicle-treated cells and are shown as the means ± SEM of triplicate experiments.

### 4.10. Cell CycleAnalysis Using Flow Cytometry

Cells were treated with a combination of SERCA inhibitors and paclitxel or sorafenib, either agent alone in RPMI-1640 medium containing 15% FBS for 40 h. The cells were then harvested by trypsinization and fixed in 70% ethanol. PTC cells were stained for total DNA using PBS containing 50 μg/mL PI (propidium iodide) and 100 μg/mL RNase I during 40 min at 37 °C. The cell cycle arrangement was examined with the FACS–CFC (Calibur Flow Cytometer, BD Biosciences, San Jose, CA, USA). The ratio of cells in the sub–G0/G1, G0/G1, S, and G2/M phases were evaluated with FlowJo v8 for MacOSX (Tree Star, Ashland, OR, USA). This experiment was repeated in triplicate and the results were averaged. The detailed protocol can be found in our previous article.

### 4.11. Immunoblot Analysis

The primary antibodies sarco/endoplasmic reticulum calcium ATPase (SERCA, 1:200, Santa Cruz Biotechnology, Santa Cruz, CA, USA, #271669), C/-EBP homologous protein (CHOP, 1:100, Santa Cruz Biotechnology, #7351), Bcl-2 (1:500, Cell Signaling Technology, Danvers, MA, USA, # 4223S), p-PERK (1:500, Abcam, Cambridge, UK, #156919), cytochrome *c* (1:100, Abcam, #90529), and β-actin (1:2000, Santa Cruz Biotechnology, #47778) were purchased and maintained overnight at 4 °C. Same quantities of protein were divided on 8–10% SDS (sodium dodecyl sulfate-polyacrylamide) gels and electro-transferred onto polyvinylidene fluoride membranes (Millipore, Bedford, MA, USA). Process of blocking was performed with 10% non–fat milk in TBS–T for 1 h at RT (room temperature). The membranes were then washed 3 times with TBS-T and probed with the corresponding secondary antibodies conjugated to HRP (horse radish peroxidase, Santa Cruz, CA, USA) at RT for 1 h. After washing, the blots were developed with ECL reagents (Pierce) and exposed using Kodak X–OMAT AR Film (Eastman Kodak, Rochester, NY, USA) for 1–5 min.

### 4.12. Human PTC Cell Xenograft

All experiments were approved by the Animal Experiment Committee of Yonsei University. YUMC-S-P1, YUMC-R-P1, -P2, and -P3 patient-derived PTC cells (4.4 × 10^6^ cells/mouse) were cultured in vitro and then injected subcutaneously into the upper left flank region of female NOD/Shi-scid, IL-2Rγ KOJic (NOG) mice. After 15 days, tumor-bearing mice were grouped randomly (*n* = 10 per group) and treated with 25 mg/kg SERCA inhibitors, thapsigargin, candidate 7 and 13 (p.o.) with 25 mg/kg paclitxel (i.p.) or 80 mg/kg sorafenib (p.o.) either alone or combination (excluded for combinations of ‘paclitaxel and sorafenib’ or SERCA inhibitors). Tumor size was measured every 3 days using calipers. Tumor volume was estimated using the following formula: L × S2/2 (L, longest diameter; S, shortest diameter). Animals were maintained under specific pathogen-free conditions. All experiments were approved by the Animal Experiment Committee of Yonsei University.

### 4.13. Statistical Analysis

For the analysis of patient data, categorical variables were expressed as frequency and proportion, whereas summary statistics (median, range) were used to report continuous data. Survival curves were generated using the Kaplan–Meier method based on the log-rank test. As this was a retrospective analysis, no formal statistical comparisons were performed. Statistical analyses were performed using GraphPad Prism 6.0 software (GraphPad Software, La Jolla, CA, USA), Microsoft Excel (Microsoft Corp, Redmond, WA, USA), and R version 2.17. One-way ANOVA was performed for the multi-group analysis, and two-tailed Student t-test was performed for the two-group analysis.

## 5. Conclusions

SERCA activation promotes influx of cytosolic free calcium into the ER, preventing cytosolic free calcium overload. It is primarily responsible for cellular resistance to genotoxic stress and apoptosis induced by anticancer drug treatment. We propose a new therapeutical approach for inducing significant tumor shrinkage, as demonstrated by using a xenograft model developed with patient-derived antineoplastic-resistant PTC cells and the novel SERCA inhibitors, candidates 7 and 13. These observations may have significant clinical implications, offering the possibility to develop novel combinatorial strategies that selectively target highly malignant cells such as drug-resistant and cancer stem-like cells.

## Figures and Tables

**Figure 1 ijms-23-10378-f001:**
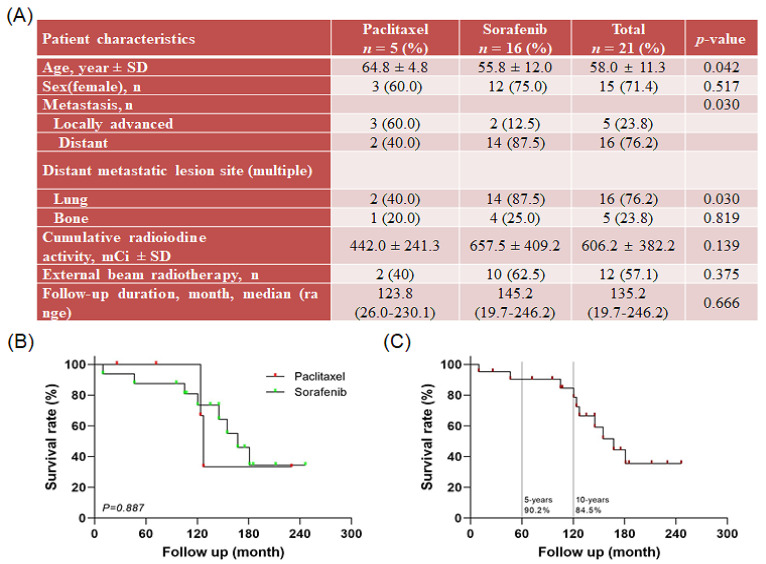
Information on patients with papillary thyroid carcinoma (PTC) after treatment with paclitaxel and sorafenib: (**A**) patient characteristics and clinical features, (**B**) overall survival rate after treatment with paclitaxel and sorafenib, and (**C**) overall survival rate of patients with refractory PTC.

**Figure 2 ijms-23-10378-f002:**
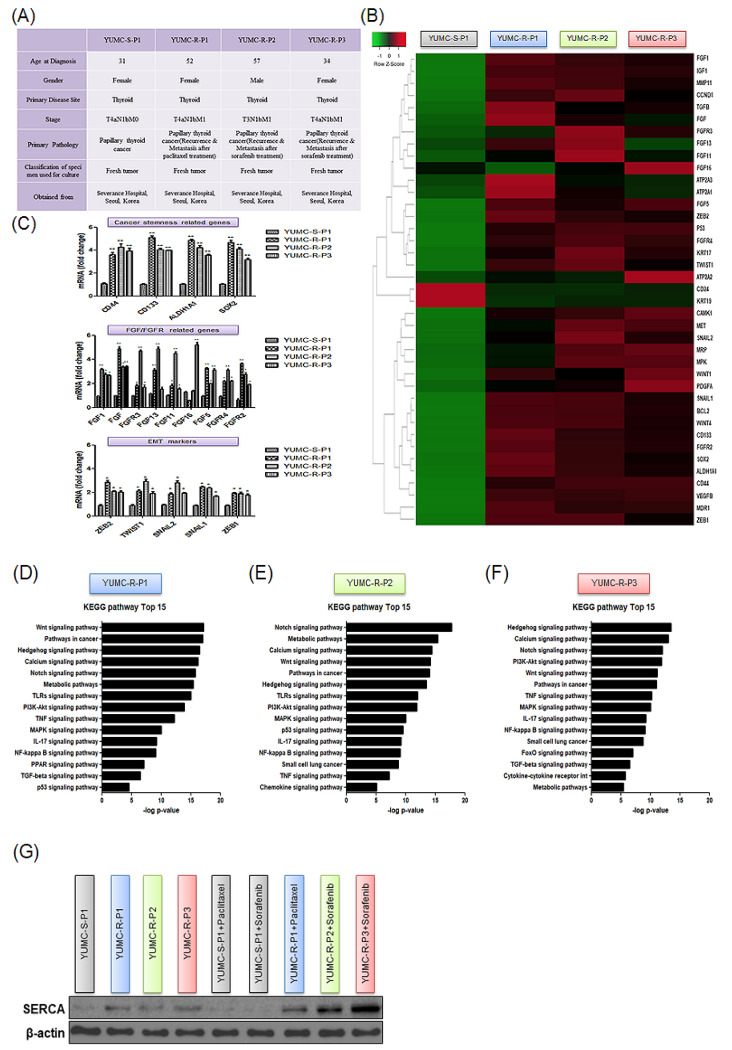
Peculiarities of all tested papillary thyroid carcinoma (PTC) cell lines. (**A**) Characteristics of patient-derived subtypes of PTC cell lines, (**B**) hierarchical clustering of genes differentially expressed; variations of gene expression profile between patient-derived antineoplastic-sensitive and paclitaxel- or sorafenib-resistant PTC cells, and (**C**) mRNA seq analysis of gene expression level for cancer stem cells (CSC) markers, fibroblast growth factor (FGF)/FGF receptor, and epithelial-mesenchymal transition (EMT) marker. Comparative analysis between antineoplastic-sensitive and paclitaxel- or sorafenib-resistant PTC cells. (**D**–**F**), bar plot illustrating the top 15 enriched pathways upregulated in patient-derived paclitaxel- or sorafenib-resistant PTC cells, YUMC-R-P1 (**D**), YUMC-R-P2 (**E**), and YUMC-R-P3 (**F**). (**G**) Comparative level of SERCA protein expression between antineoplastic-sensitive and paclitaxel- or sorafenib-resistant PTC cells under untreated and paclitaxel- or sorafenib-treated conditions. * *p* < 0.05 vs. antineoplastic-sensitive PTC cells, YUMC-S-P1, ** *p* < 0.01 vs. antineoplastic-sensitive PTC cells, YUMC-S-P1.

**Figure 3 ijms-23-10378-f003:**
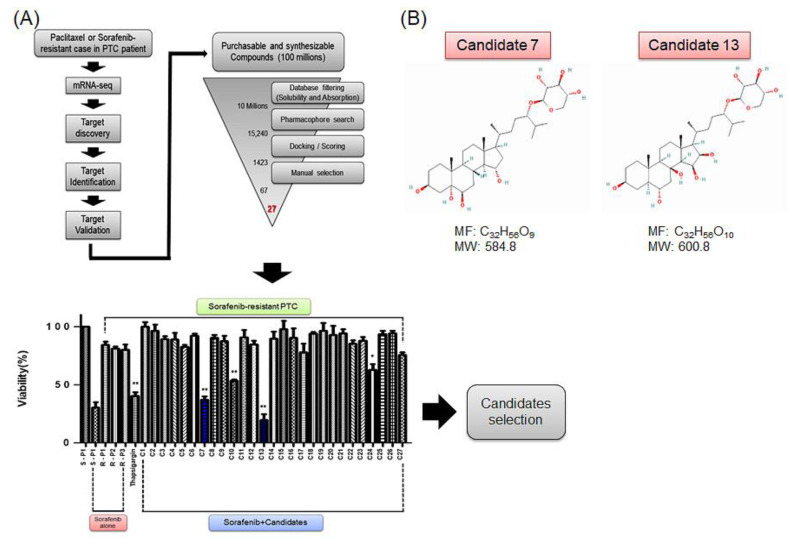
Scheme of study design and in silico screening using the evolutionary chemical binding similarity (ECBS) method for discovery of novel SERCA inhibitors. (**A**) General categorization and selection of novel SERCA inhibitors through ECBS methods. (**B**) Chemical structures and information of novel SERCA inhibitors, candidates 7 and 13. * *p* < 0.05. ** *p* < 0.01.

**Figure 4 ijms-23-10378-f004:**
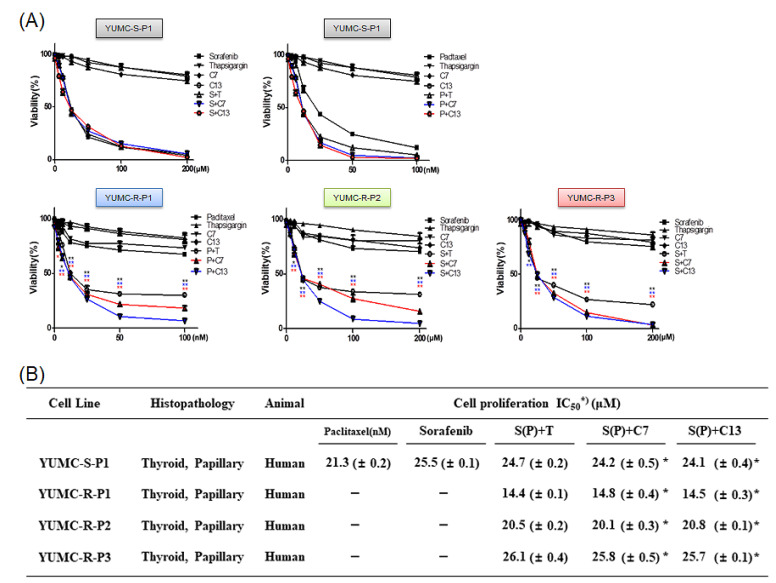
Anticancer efficacy of combination therapy using novel SERCA inhibitors (thapsigargin, positive control of SERCA inhibitor; candidate 7 and 13, novel SERCA inhibitors) and anticancer drugs (paclitxel or sorafenib). Comparative efficacy on antineoplastic-sensitive and paclitaxel- or sorafenib-resistant PTC cells. Cell viability of antineoplastic-sensitive (**A**, YUMC-S-P1, top) and paclitaxel- or sorafenib-resistant (**A**). YUMC-R-P1, -R-P2, and -R-P3 (bottom) PTC cells after exposure to SERCA inhibitors and paclitaxel or sorafenib alone or in combination. Points represent the mean percentage of the values determined in the solvent-treated control. All experiments were performed in triplicate (minimum). Data represent mean ± standard deviation. * *p* < 0.05 and ** *p* < 0.01 versus control. *, ** paclitaxel or sorafenib + thapsigargin versus control, *, ** paclitaxel or sorafenib + candidate 7 versus control, *, ** paclitaxel or sorafenib + candidate 13 versus control. (**B**). IC_50_ values of the combination of SERCA inhibitors with paclitaxel or sorafenib in antineoplastic-sensitive and paclitaxel- or sorafenib-resistant PTC cells. Each data point corresponds to the mean of three separate MTT assays, carried out in triplicate. SEM, standard error of the mean; MTT, 3-(4,5-dimethylthiazol-2-yl)-2,5-diphenyltetrazolium bromide; IC_50_, half-maximal inhibitory concentration. The asterisk indicates lowest half-maximal inhibitory concentration.

**Figure 5 ijms-23-10378-f005:**
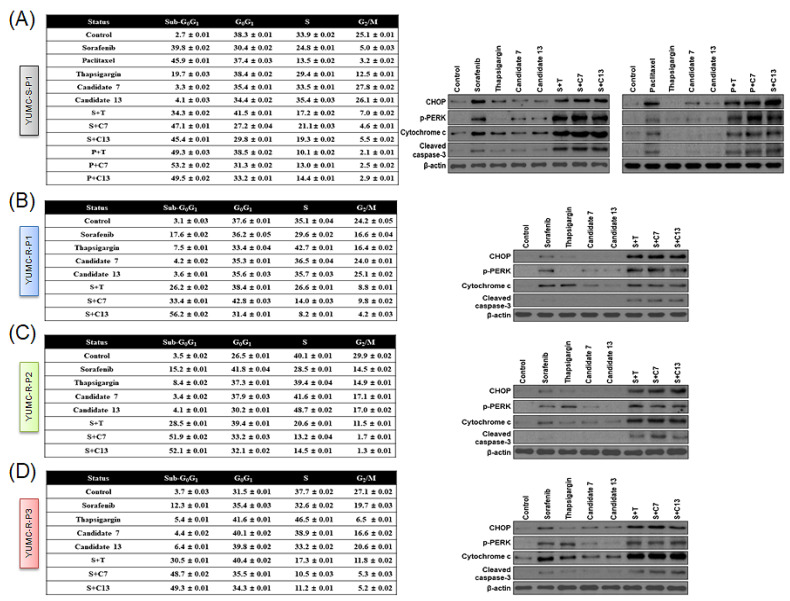
Level of apoptosis evaluated through quantitation of DNA content with propidium iodide and immunoblot analysis. Flow cytometry (**A**–**D**, **left**) and immunoblot analysis (**A**–**D**, **right**) of patient-derived antineoplastic-sensitive (YUMC-S-P1) and paclitaxel- or sorafenib-resistant PTC cells (YUMC-R-P1, -P2, and -P3). (**A**–**D**, **left**) Cells were exposed to the indicated SERCA inhibitors, paclitxel or sorafenib alone or in combination, harvested, and stained with propidium iodide before analysis by flow cytometry, using FlowJo version 8. (**A**–**D**, **right**), Immunoblot analysis—comparative levels of expression of CHOP (endoplasmic reticulum stress marker), cytochrome *c*, and cleaved caspase-3 (apoptosis-related marker) between antineoplastic-sensitive (YUMC-S-P1) and paclitaxel- or sorafenib-resistant PTC cells (YUMC-R-P1, -P2, and -P3).

**Figure 6 ijms-23-10378-f006:**
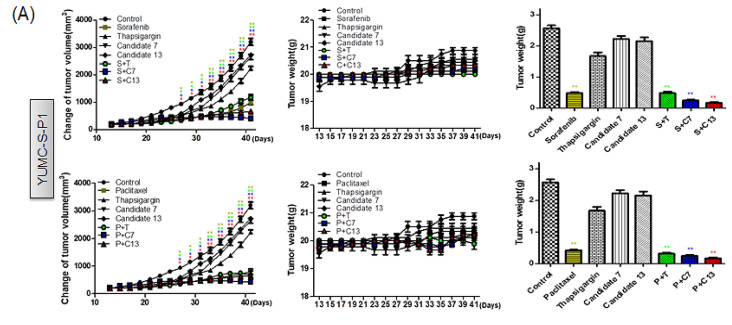
Combination of novel SERCA inhibitors with paclitaxel or sorafenib has the highest anticancer efficiency, inducing a higher tumor shrinkage in patient-derived antineoplastic-resistant PTC cells, YUMC-R-P1, -P2, and -P3 in vivo than in sensitive cells (YUMC-S-P1). (**A**–**D**, **left**). Changes in the tumor volume; (**A**–**D**, **right**). The weight of the resected tumor; (**A**–**D**, **middle**). Changes in whole body weight (each group, *n* = 10). Tumors were evaluated in NOD/Shi-scid, IL-2Rγ KOJic (NOG) mice and animals were treated with paclitaxel or sorafenib combined with novel SERCA inhibitors, candidates 7 or 13, or with each agent alone. Data represent the mean ± standard error of the mean. * *p* < 0.05 and ** *p* < 0.01, compared with control. *,** paclitaxel or sorafenib + thapsigargin versus control, *,** paclitaxel or sorafenib + candidate 7 versus control, *,** paclitaxel or sorafenib + candidate 13 versus control.

## Data Availability

The data presented in this study are available upon reasonable request from the corresponding author.

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
