# Peer review of "Potential Therapeutic Agents against Paclitaxel—And Sorafenib-Resistant Papillary Thyroid Carcinoma"

_ijms, 2022, doi:10.3390/ijms231810378_

Round 1

Reviewer 1 Report

Kim and colleagues reported that combined treatment with a SERCA inhibitor can reverse resistance to chemotherapy (paclitaxel) or tyrosine kinase inhibitors (sorafenib) in thyroid cancer cells and induce ER stress-mediated apoptosis. The study design is very similar to that of a previous study from the same group (ref #31). The main difference is in the type of tumor. However, thyroid cancer (particularly differentiated thyroid cancer) has many unique characteristics that are different to other cancers. The manuscript needs to be thoroughly revised by a clinician majoring in endocrinology or surgical endocrinology.

#1. Abstract, introduction, and discussion were poorly written and poorly referenced, e.g. "these refractory PTCs are commonly characterized by anaplastic transformation of the tumor cells." This is incorrect. PTCs may become radioiodine-refractory and less differentiated, but morphologically, they do not undergo anaplastic transformation.
#2. It should be emphasized that paclitaxel is not a standard treatment for thyroid cancer.
#3. Four cell lines were grouped as 'patient-derived drug-sensitive/resistant PTC cell lines'. However, Patient 3 (YUMC-R-P2) had poorly differentiated thyroid carcinoma (not PTC). In Figure 2A, it was described as PTC. Needs to be clarified.
#4. YUMC-R-P1~P3 were more aggressive than YUMC-S-P1. Please define 'more aggressive.'
#5. To claim that SERCA inhibitors promote ER stress-mediated apoptosis, in addition to CHOP, other markers (such as PERK/ATF6/IRE1) may be evaluated. Do ER stress inhibitors abolish the effects of SERCA inhibitors?
#6. Were novel inhibitors (candidates 7 or 13) more effective than classic inhibitors such as thapsigargin? If not, the significance of these novel inhibitors is of questionable importance.
#7. Were RNA-seq data stored in a public repository? Were there differences in the expression of thyroid differentiation genes (e.g. NIS, TSHR)?
#8. Figure 6 is missing from the manuscript.
#9. The placement of figures is confusing (e.g. Fig 3 misplaced next to Fig 5 legend).
#10. Please change Figure 4A to a color plot with different colors within each group. This makes it more readable.
#11. Figure 5 (cell cycle distribution) may be expressed as plots.
#12. "Immunohistochemistry results were subjected to one-way analysis of variance, followed by Bonferroni post-hoc test." I have not found any immunohistochemistry in this study.
#13. Page 3: [32296030] means?
#14. References are poorly formatted with lots of incorrect lower-case letters. Ref #45 is irrelevant to the text.

Author Response

Kim and colleagues reported that combined treatment with a SERCA inhibitor can reverse resistance to chemotherapy (paclitaxel) or tyrosine kinase inhibitors (sorafenib) in thyroid cancer cells and induce ER stress-mediated apoptosis. The study design is very similar to that of a previous study from the same group (ref #31). The main difference is in the type of tumor. However, thyroid cancer (particularly differentiated thyroid cancer) has many unique characteristics that are different to other cancers. The manuscript needs to be thoroughly revised by a clinician majoring in endocrinology or surgical endocrinology.

> Reply: I don't know how to thank you enough for reviewing our manuscript. I agree with you completely and follow your professional opinion. Thank you again for your review. I hope you are always healthy and happy!!

#1. Abstract, introduction, and discussion were poorly written and poorly referenced, e.g. "these refractory PTCs are commonly characterized by anaplastic transformation of the tumor cells." This is incorrect. PTCs may become radioiodine-refractory and less differentiated, but morphologically, they do not undergo anaplastic transformation.

> Reply: Thank you for your comment. I follow your professional opinion, I have corrected writing and referencing in whole manuscript. Thank you again for your expert advice.

#2. It should be emphasized that paclitaxel is not a standard treatment for thyroid cancer.

> Reply: I agree with you completely your professional opinion. Yes, your advice is correct. However, we obtained patient-derived drug resistant PTC case was only paclitaxel or sorafenib resistant case. As you said, paclitaxel is not a standard treatment for PTC, but it has been used in actual clinical trials due to problems such as the National Health Insurance (before approval of sorafenib or levatinib. This patient underwent paclitaxel after which the disease progression was confirmed in the anti-cancer drug response evaluation. Cancer recurrence and metastasis were caused after paclitaxel prescribed. Consequently we researched in case of paclitaxel or sorafenib resistant PTC. In this article, we tried to showing SERCA is one of critical regulator for the survival under severe ER stress like treatment of paclitaxel or sorafenib.

#3. Four cell lines were grouped as 'patient-derived drug-sensitive/resistant PTC cell lines'. However, Patient 3 (YUMC-R-P2) had poorly differentiated thyroid carcinoma (not PTC). In Figure 2A, it was described as PTC. Needs to be clarified.

> Reply: Thank you for your comment. The above patient was identified as PTC in the first bilateral total thyroidectomy and was also confirmed in kidney and lung specimens. The samples used in subsequent experiments were obtained during the last regional LN dissection and were changed by PTC as poorly differentiated thyroid carcinoma. This is obviously an error in our description, and we updated the modifications. Corrected sentences were added in Materials and Methods, ‘4.2.3. Patient 3’.

#4. YUMC-R-P1~P3 were more aggressive than YUMC-S-P1. Please define 'more aggressive.'

> Reply: I follow your professional opinion, I have corrected ‘more aggressive’ to ‘more aggressive (metastasis or recurrent)’.

#5. To claim that SERCA inhibitors promote ER stress-mediated apoptosis, in addition to CHOP, other markers (such as PERK/ATF6/IRE1) may be evaluated. Do ER stress inhibitors abolish the effects of SERCA inhibitors?

> Reply: I agree with you completely and follow your professional opinion. I have made the suggested correction in Figure 5A-D. In ours humble opinion, SERCA could be specific regulator for survival by overloaded cytosolic free calcium restoration to ER under severe ER stress like anti-cancer drug treatment in drug resistant PTC cells. If non-severe ER stress conditions by ER stress inhibitors, role of the SERCA might be no significantly influenced to calcium homeostasis.

#6. Were novel inhibitors (candidates 7 or 13) more effective than classic inhibitors such as thapsigargin? If not, the significance of these novel inhibitors is of questionable importance.

> Reply: Thank you for your comment. Novel SERCA inhibitors, candidates 7 and 13 were more effective than classic inhibitors such as thapsigargin in dose dependant manners. Moreover, these SERCA inhibitors were showed no significantly influenced cardiovascular and muscular diseases through in vivo test. While anti-cancer effect of the thapsigargin is well known, side effects are also known. In ours humble opinion, effective anti-cancer drug is so important, but safe anti-cancer drug will also be imprtant.

#7. Were RNA-seq data stored in a public repository? Were there differences in the expression of thyroid differentiation genes (e.g. NIS, TSHR)?

> Reply: SLC5A5 (NIS) RNA expressions were significantly decreased in paclitaxel or sorafenib resistant PTC cells compare than paclitaxel or sorafenib sensitive PTC cells. The expression of thyroid differentiation related genes in anti-cancer resistant PTC cells will be investigated our next artlcle. And i am very sorry, several informations were patent pending, please understand that the informations cannot be disclosed.

#8. Figure 6 is missing from the manuscript.

> Reply: We are very sorry for the confusion caused a missing of the figure 6. There seems to be a miss communications with the editor. Thank you for your comment. I have made this correction.

#9. The placement of figures is confusing (e.g. Fig 3 misplaced next to Fig 5 legend).

> Reply: We are so sorry for the confusion caused. There seems to be a miss communications with the editor. Thank you for your comment. I have made this correction.

#10. Please change Figure 4A to a color plot with different colors within each group. This makes it more readable.

> Reply: Thank you for your comment. I follow your professional opinion, Figure 4A was corrected more readable. Moreover information of color plot was added in figure legend. Thank you again for your expert comments

.
#11. Figure 5 (cell cycle distribution) may be expressed as plots.

> Reply: Thank you for your comment. I agree with your professional opinion. However, in ours humble opinion, because figure 5B was consisted to 20 cases of plots, we afraid of causing confusion among readers. For this reason, we tried to express it as siple as possible, so please understand.  

#12. "Immunohistochemistry results were subjected to one-way analysis of variance, followed by Bonferroni post-hoc test." I have not found any immunohistochemistry in this study.

> Reply: We are very sorry for the confusion caused a faulty typing. I have made the suggested correction.

#13. Page 3: [32296030] means?

> Reply: Thank you for your comment. There was mistyping and made this correction.

#14. References are poorly formatted with lots of incorrect lower-case letters. Ref #45 is irrelevant to the text.

> Reply: I agree with you completely and follow your professional opinion. I have made the suggested correction.

Reviewer 2 Report

The manuscript "Potential therapeutic agents against paclitaxel- and sorafenib-resistant papillary thyroid carcinoma" is focused on the development of a new clinical approach for the treatment of therapy-refractory papillary thyroid cancer (PTC) based on the use of inhibitors of sarco/endoplasmic reticulum calcium-ATPase (SERCA) in combination with paclitaxel or sorafenib. This work showed that treatments with novel SERCA inhibitors (compounds 7 and 13) significantly reduced tumor growth in a patient-derived xenograft tumor model developed using paclitaxel- or sorafenib-resistant PTC cells.

This manuscript is well written and might be interesting for oncologists and basic researchers working in the cancer biology field. The significance of the studied issues and the study's novelty are substantiated in the Introduction. However, data from the 2.5 Results section are not presented in Figure 5. In general, the presented results support the conclusions, but an absence of data on apoptosis analysis makes it impossible to evaluate this manuscript. I have the following comments on the manuscript.

 1. There is no Figure 5 in the manuscript, which makes it difficult to understand and evaluate the presented study.

2. Figure 2C presents a comparative mRNA seq analysis of gene expression level for cancer stem cells (CSC) markers, fibroblast growth factor (FGF)/FGF receptor, and epithelial-mesenchymal transition (EMT) marker between antineoplastic-sensitive and paclitaxel- or sorafenib-resistant PTC cells. This group presentation of data is incorrect and needs to be corrected. Data should be reported for each gene, not for a group.

3. Methods are not described sufficiently, referencing a previous publication. It is necessary to add information to the description of all methods in a short form.

4. It is necessary to increase the resolution for the Figures, as the information on some graphs and tables is challenging to read. It also needs to correct the location of the Figures and legends in the text.

Author Response

The manuscript "Potential therapeutic agents against paclitaxel- and sorafenib-resistant papillary thyroid carcinoma" is focused on the development of a new clinical approach for the treatment of therapy-refractory papillary thyroid cancer (PTC) based on the use of inhibitors of sarco/endoplasmic reticulum calcium-ATPase (SERCA) in combination with paclitaxel or sorafenib. This work showed that treatments with novel SERCA inhibitors (compounds 7 and 13) significantly reduced tumor growth in a patient-derived xenograft tumor model developed using paclitaxel- or sorafenib-resistant PTC cells.

This manuscript is well written and might be interesting for oncologists and basic researchers working in the cancer biology field. The significance of the studied issues and the study's novelty are substantiated in the Introduction. However, data from the 2.5 Results section are not presented in Figure 5. In general, the presented results support the conclusions, but an absence of data on apoptosis analysis makes it impossible to evaluate this manuscript. I have the following comments on the manuscript.

> Reply: I don't know how to thank you enough for reviewing our manuscript. I agree with you completely and follow your professional opinion. Thank you again for your review. I hope you are always healthy and happy!!

  1. There is no Figure 5 in the manuscript, which makes it difficult to understand and evaluate the presented study.

> Reply: We are very sorry for the confusion caused a missing of the figure 5. There seems to be a miss communications with the editor. Thank you for your comment. I have made this correction.

  1. Figure 2C presents a comparative mRNA seq analysis of gene expression level for cancer stem cells (CSC) markers, fibroblast growth factor (FGF)/FGF receptor, and epithelial-mesenchymal transition (EMT) marker between antineoplastic-sensitive and paclitaxel- or sorafenib-resistant PTC cells. This group presentation of data is incorrect and needs to be corrected. Data should be reported for each gene, not for a group.

> Reply: I follow your professional opinion, I have made the suggested correction. Figure 2C was corrected for each gene, not for a group. Thank you again for your expert opinion.

  1. Methods are not described sufficiently, referencing a previous publication. It is necessary to add information to the description of all methods in a short form.

> Reply: Thank you for your comment. I follow your professional opinion, corrected whole ‘M&M’ (detail information included) was added in ‘Materials and methods’.

  1. It is necessary to increase the resolution for the Figures, as the information on some graphs and tables is challenging to read. It also needs to correct the location of the Figures and legends in the text.

> Reply: We are very sorry for the confusion caused miss-matching of figures and legends. Whole article was corrected follow your professional opinion.

Round 2

Reviewer 1 Report

The authors have addressed all my concerns.

Reviewer 2 Report

The revised manuscript has been significantly improved: necessary changes have been made to the text and figures, and the quality of "Materials and Methods" has been improved. The authors corrected and added data on comparative mRNA analysis of gene expression levels in Figure 2C.